# Self-report assessment of Positive Appraisal Style (PAS): Development of a process-focused and a content-focused questionnaire for use in mental health and resilience research

Papoula Petri-Romão[1]*, Haakon Engen[2], Anna Rupanova[1,2], Lara Puhlmann[1], Matthias Zerban[2], Rebecca J. Neumann[3], Aliaksandr Malyshau[3], Kira F. Ahrens[3], Anita Schick[4], Bianca Kollmann[1,5], Michèle Wessa[1,6], Henrik Walker[7,8], Michael M. Plichta[3], Andreas Reif[3], Andrea Chmitorz[9], Oliver Tuescher[1,5], Ulrike Basten[10], Raffael Kalisch[1,2]*

1 Leibniz Institute for Resilience Research (LIR), Mainz, Germany, 2 Neuroimaging Center (NIC), Focus Program Translational Neuroscience (FTN), Johannes Gutenberg University Medical Center, Mainz, Germany, 3 Department of Psychiatry, University Hospital, Psychosomatic Medicine and Psychotherapy, Goethe University Frankfurt, Frankfurt, Germany, 4 Medical Faculty Mannheim, Department of Public Mental Health, Central Institute of Mental Health (CIMH), Heidelberg University, Mannheim, Germany, 5 Department of Psychiatry and Psychotherapy, Johannes Gutenberg University Medical Center, Mainz, Germany, 6 Department of Clinical Psychology and Neuropsychology, Institute of Psychology, Johannes Gutenberg University, Mainz, Germany, 7 Department of Psychiatry & Psychotherapy, Division of Mind and Brain Research, Campus Charité Mitte, Charité—Universitätsmedizin Berlin, Berlin, Germany, 8 Berlin School of Mind and Brain, Humboldt-Universität zu Berlin, Berlin, Germany, 9 Faculty of Social Work, Health Care and Nursing, Esslingen University of Applied Sciences, Esslingen, Germany, 10 Department of Psychology, University of Kaiserslautern-Landau, Landau, Germany

* papoula.petri-romao@lir-mainz.de (PP-R); rkalisch@uni-mainz.de (RK)

## Abstract

Positive Appraisal Style Theory of Resilience posits that a person's general style of evaluating stressors plays a central role in mental health and resilience. Specifically, a tendency to appraise stressors positively (positive appraisal style; PAS) is theorized to be protective of mental health and thus a key resilience factor. To this date no measures of PAS exist. Here, we present two scales that measure perceived positive appraisal style, one focusing on cognitive processes that lead to positive appraisals in stressful situations (PASS-process), and the other focusing on the appraisal contents (PASS-content). For PASS-process, the items of the existing questionnaires Brief COPE and CERQ-short were analyzed in exploratory and confirmatory factor analyses (EFA, CFA) in independent samples (N = 1157 and N = 1704). The resulting 10-item questionnaire was internally consistent (α = .78, 95% CI [.86, .87]) and showed good convergent and discriminant validity in comparisons with self-report measures of trait optimism, neuroticism, urgency, and spontaneity. For PASS-content, a newly generated item pool of 29 items across stressor appraisal content dimensions (probability, magnitude, and coping potential) were subjected to EFA and CFA in two independent samples (N = 1174 and N = 1611). The resulting 14-item scale showed good internal consistency (α = .87, 95% CI [.86, .87]), as well as good convergent and discriminant validity within

**Data Availability Statement:** Data can be found on the osf project https://osf.io/kt2h8/.

**Funding:** This project has received funding from the European Union's Horizon 2020 (https://research-and-innovation.ec.europa.eu/funding/funding-opportunities/funding-programmes-and-open-calls/horizon-2020_en) research and innovation program under Grant Agreement numbers 777084 (RK; DynaMORE project) and 101016127 (OT, RK; RESPOND project), from the German Research Foundation (AR; DFG CRC 1193, subproject Z03; https://www.dfg.de/en/), from the Stiftung Rheinland-Pfalz für Innovation (RK; MARP program, No 961-386261/1080; https://mwg.rlp.de/de/startseite/), and from the Ministry of Science of the state of Rhineland-Palatinate (RK; Ministerium für Wissenschaft, Weiterbildung und Kultur, Rheinland-Pfalz,DRZ program; https://mwg.rlp.de/de/startseite/). The Gutenberg Brain Study (OT) has received funding from Focus Program Translational Neuroscience (FTN) from the Ministry of Science of the state of Rhineland-Palatinate and the University Medical Center Mainz (Ministerium für Wissenschaft, Weiterbildung und Kultur, Rheinland-Pfalz and Universitätsmedizin Mainz https://www.blogs.uni-mainz.de/ftn-eng/).

**Competing interests:** RK has received advisory honoraria for consultancy from the commercial source JoyVentures, Herzlia, Israel. This does not interfere with the full and objective conduct and presentation of the research and does not alter our adherence to PLOS ONE policies on sharing data and materials.

the nomological network. The two scales are a new and reliable way to assess self-perceived positive appraisal style in large-scale studies, which could offer key insights into mechanisms of resilience.

## Introduction

Psychological resilience has been defined as the maintenance or quick recovery of mental health during and after exposure to stressors, or adversity [1]. This conceptualization of resilience reflects the observation that even severe adversity does not always lead to lasting stress-related dysfunction [2]. Resilience is thus defined as an outcome. Resilience research, accordingly, is concerned with identifying the mechanisms that protect against the development of mental health problems in stressor-exposed individuals and with harnessing this knowledge for the prevention, rather than the treatment, of dysfunction [3]. Individual differences, such as one's environmental circumstances, one's personality as well as cognitive processes or biological factors, play a role in achieving resilient outcomes [4,5]. Positive Appraisal Style Theory of Resilience (PASTOR) [6] positions positive appraisal style as one of these individual differences. To test this resilience theory in large-scale studies, succinct self-report questionnaires need to be developed [7].

### Positive appraisal style

PASTOR theory relies on the concept of appraisal, which is the function of analyzing a stimulus or situation with respect to its meaning for the organism and, on this basis, determining the emotional reaction to the stimulus or situation. Stress reactions result from an appraisal of a stimulus/situation, that is, of a potential stressor, as a threat to one's goals or needs. Stressor appraisal occurs along the three major threat appraisal dimensions of *threat magnitude* or *cost*, *threat probability*, and *coping potential* [6]. Appraisal may be implemented via a heterogeneous set of cognitive processes. Some of these processes may be unconscious, non-verbal, and implicit, while others make use of verbal and conscious-explicit mental operations [6]. Appraisal determines the values attributed to a stressor on the aforementioned dimensions. In positive appraisal these values are set to levels that realistically the threat or even slightly underestimate it see Fig 1 for a schematic overview). That is, positive appraisal avoids catastrophizing (*magnitude/cost* dimension), pessimism (*probability* dimension), and perceived helplessness (*coping* dimension). In this sense, positive appraisal is primarily a non-negative way of appraising stressors. At the same time, positive appraisal avoids unrealistically positive (delusional) threat perceptions that might lead to trivialization, blind optimism, or extreme over-confidence.

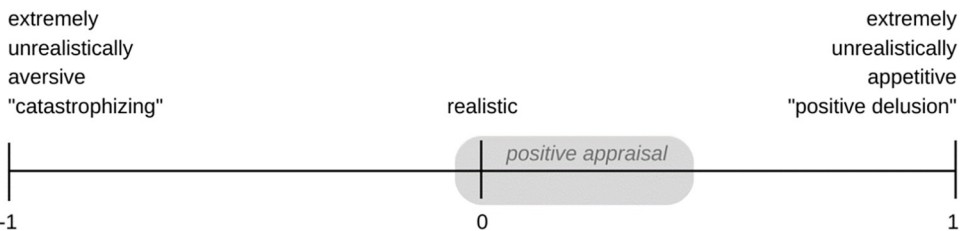

**Fig 1. Schematic illustration of the range of appraisals constituting 'positive appraisal'.** PAS (marked in gray) according to Positive Appraisal Style Theory of Resilience (PASTOR). Adapted from Kalisch and colleagues [8] © Cambridge University Press 2015, reproduced with permission.

PASTOR assumes that an individual usually appraises similar situations in a similar fashion, and therefore can be characterized by their typical appraisal tendencies ('appraisal style') [6]. A *negative appraisal style* (NAS) consists in a propensity to overestimate the aversive consequences and the probability of challenging situations and to underestimate one's coping potential. A NAS results in consistent over-reactions to perceived threats, that is, unnecessarily strong, long, or frequent stress reactions. Insofar as stress reactions consume resources (energy, time, cognitive capacity etc.), frequently generating unnecessary reactions can lead to resource depletion and allostatic load effects and eventually increase the likelihood of developing stress-related mental dysfunctions when confronted with substantial adversity [9]. A *positive appraisal style* (PAS), by contrast, is defined as the absence of such negative biases, but also by the absence of delusional, overly positive appraisal tendencies. As a result, individuals exhibiting PAS will typically generate appropriate stress responses to a given situation as necessary. The mild tendency to under-react will increase the possibilities to replenish and build resources and to learn from exploration and encounters with new situations [8]. Consequently, the likelihood of developing a mental disorder in adverse circumstances is decreased relative to individuals with a negative evaluative style. For this reason, PASTOR claims that PAS is a key resilience factor [8].

PAS is considered a relatively stable, yet malleable individual difference that may change as a function of experience, reflection, or instruction [6,8]. In addition, PAS would theoretically reduce allostatic overload by reducing exaggerated stress responses, which in turn can prevent pathological processes at a very early stage before any specific body or brain system can be lastingly affected. This positions PAS a potential protective factor against many or most potential stress-related dysfunctions.

These theoretical considerations suggest that a PAS questionnaire could be of high benefit for mental health and resilience research. In the present paper, we describe a PAS instrument development for the general population.

### Appraisal processes versus appraisal contents

There is an important conceptual distinction in appraisal theory between appraisal *processes* and appraisal *contents*. Appraisal *processes* are the conscious or non-conscious mental operations that generate appraisal outcomes. Appraisal *contents*, on the other hand, are the mental representations of the meaning of a stimulus or situation for somebody's needs and goals [8]. Importantly, appraisal *contents* are the outcome of appraisal *processes*.

**Positive appraisal processes.**   Positive appraisals are generated in demanding or threatening situations by a range of neural/cognitive processes. PASTOR classifies these processes into three broad classes: positive situation classification (a), positive reappraisal (b), and inhibition (c) [6]. (a) In situations that are only mildly aversive, positive appraisals are generated more or less automatically through a process of positive *situation classification* (PASTOR process class 1). This process would be based on easily accessible or highly generalized previous experiences where such aversive situations have been handled successfully, or on strong cultural tendencies to view a situation as non-threatening. (b) The second class of processes would be required when a situation is sufficiently aversive to cause initial and more or less unavoidable negative appraisals. In this event, a *reappraisal* of the situation is necessary to eventually appraise the situation positively (process class 2). This re-evaluation can vary in the extent to which it is conscious, volitional, effortful, and verbal. Reappraisal can consist in discovering new positive aspects of a situation, in weighting positive aspects of a situation more positively than initially, in re-construing aspects of a situation initially perceived as negative in a positive fashion, in distancing from a situation, or in considering a situation acceptable. It can in turn lead to

safety learning and the extinction of previous aversive stimulus-response relationships [8]. It must not necessarily lead to the generation of a positive emotional state but may merely reduce the extent of a negative reaction. Accordingly, PASTOR's definition of reappraisal processes is broad and encompasses any change in appraisals fulfilling the described functions, including potentially non-conscious and non-verbal processes [6]. (c) The third and last process class (class 3) is the inhibition of competing negative appraisals and negative emotional responses in progress and is considered to support reappraisal [6].

Assessing appraisal processes via self-report is challenging, as it requires process awareness and consequently could be impossible for processes situated outside consciousness [8]. This may be the case for many processes contributing to the more or less automatic situation classification (in class 1) but also for many implicit reappraisal and inhibition processes (in classes 2 and 3). Any appraisal process self-report instrument for PAS will therefore only cover a subset of relevant processes, namely the more 'cognitive' appraisal and reappraisal processes, many of which may be subsumed under the umbrella term of 'positive cognitive reappraisal' [10]. It will inevitably suffer from limited validity due to content under-representation [11]. Another limitation is that it may be questionable to ask a participant to realistically report on their ability, or capacity, to use a given process (process effectiveness), an aspect of process assessment that may be better performed in the research laboratory with the help of a controlled experimental (re)appraisal task that includes success measures [10,12]. Most instruments used to assess thinking processes therefore focus on participants' tendency to use a given process in specific situations (process frequency) [13–15]. In sum, self-report measures of appraisal processes should primarily target the (self-perceived) frequency of conscious positive (re)appraisal processes.

It is unclear whether different conscious positive appraisal and reappraisal processes are used independently. That is, whether the frequent use of one reappraisal tactic, for instance positive reinterpretation, would be associated with frequent use of other tactics, such as putting into perspective, distancing, or acceptance. In existing questionnaires, corresponding subscales are often well correlated [13,14], suggesting there might be a common latent dimension that reflects a positive cognitive (re)appraisal tendency.

**Positive appraisal contents.**    As said, PASTOR posits that potential stressors are appraised on three major content dimensions (threat probability, threat magnitude, and coping potential) [6]. Similar to positive appraisal processes, appraisal contents may be non-conscious or conscious (thoughts) and therefore only partly accessible to self-report. PASTOR considers these content dimensions to be at least partly unrelated. One can be very pessimistic about the materialization of a potential threat (such as a job loss after one has insulted one's boss) but at the same time estimate one's coping potential for a worst-case scenario as high (e.g., because one has good financial reserves). One can also find that it would be horrible if a threat materialized (such as if the plane one has boarded crashed) but at the same time consider this scenario highly improbable (as is appropriate for unconditional plane crash probability estimates in modern aviation). In both examples, the resulting overall stress reaction would be rather mild despite one threat appraisal dimension (probability in the first example, magnitude/cost in the second example) being assigned a high value. These examples illustrate the principled independence of the three threat appraisal dimensions. In extension to individual styles or tendencies of appraisal on these dimensions, this implies that an overall positive appraisal style does not necessarily require that the individual usually generates positive appraisal contents on all three dimensions. Theoretically, a pessimist may generally consider themselves good at coping, and a habitual catastrophizer may still be an optimist. From this point of view, negative tendencies on one dimension can thus be compensated to some extent by positive tendencies on another dimension, such that overall appraisal style becomes a sum game.

Nonetheless, it remains unclear whether dimensional appraisal tendencies are truly independent. There is some support that the dimensions are in fact related, as dispositional optimism scores are known to correlate with scores on instruments assessing the coping dimension of appraisal, such as control beliefs, powerlessness, self-efficacy, sense of mastery, or autonomy, as well as the cost/magnitude dimension, such as catastrophizing [16–19].

*Developing a PAS measurement.* Taken together, the distinction between appraisal processes and contents and the associated methodological considerations open two different roads towards assessing PAS with self-report instruments: (1) assessing the frequency of various positive cognitive appraisal and reappraisal processes in stressful situations (process-focused approach), and (2) assessing the positive appraisal contents that individuals typically generate in response to stressors on the three key threat appraisal dimensions (content-focused approach). A content-focused measure would then be a more direct measure of PAS, as it measures positive appraisals ignorant of the underlying processes. In contrast, a process-focused measure assesses the self-perceived frequency of consciously accessible processes. For both types of instruments, the cited empirical observations raise the strategic question of whether the instrument should aim at covering different types of appraisal processes or different appraisal content dimensions, respectively, as a collection of separate, more or less independent subscales for each process type or content dimension. Or whether, alternatively, one should strive to obtain an instrument with a simple factor structure (perhaps a single factor) and high internal consistency. The latter solution would have psychometric advantages but may sacrifice detail, for instance by making it difficult to determine whether a pessimist could have strong coping beliefs (second instrument type).

**Existing measures relevant for positive appraisal style.** *Processes.* Existing measures serving to assess the frequency of positive cognitive appraisal and reappraisal processes in the general population include the Emotion Regulation Questionnaire (ERQ)[15], the COPE [13], and the Cognitive Emotion Regulation Questionnaire (CERQ)[20], the latter two also available in popular short versions (Brief COPE; CERQ-short) [14,21]. None of these questionnaires exclusively investigate positive cognitive (re)appraisal. For instance, COPE and CERQ also contain subscales that reflect non-appraisal-based cognitive processes, such as distraction; these questionnaires also measure negative appraisal processes, such as self-blame; and the ERQ and the COPE also feature behaviors that people show in challenging situations, such as expressive suppression (ERQ), active coping, or substance use (COPE). The largest collection of potentially relevant items can be found in the COPE and CERQ. Subscales which can be relatively readily classified as targeting positive cognitive (re)appraisal processes in the broad definition of PASTOR are *Positive reframing* (COPE), *Positive reappraisal* (CERQ), *Putting into perspective* (CERQ), and *Acceptance* (COPE and CERQ). A limitation is that these subscales also contain items that, upon inspection of the employed wording, could better be classified as assessing appraisal contents rather than processes, or as being ambiguous with respect to this conceptual distinction. Another ambiguity present specifically in the COPE is that some subscales do not clearly distinguish between cognitive versus behavioral processes. This applies to *Turning to religion* (where an item such as prayer could be seen as a cognitive process or as a behavior) and *Behavioral disengagement* (which has aspects of cognitive acceptance). Finally, for some items, notably in the *Rumination*, *Humor*, and *Other-blame* subscales, it is not clear if they should be considered as positive and presumably adaptive approaches to potentially threatening situations. So, rumination, typically studied for negative contents, can be used to dwell on positive aspects of ambiguous situations [22] and to thereby drive the overall situation appraisal towards the positive. Humor, on the other hand, appears to be a positive appraisal strategy but could also be seen as a distraction, or cognitive avoidance, strategy. Other-blame may also serve to avoid cognitive confrontation with potentially uncomfortable information but could also be seen as positive reappraisal of ego threats, up-regulating self-esteem.

Taken together, existing measures cannot be used as such to measure positive appraisal processes but may be useful as sources of potentially relevant items.

*Contents.* The literature already offers various instruments assessing appraisal contents on the three threat appraisal dimensions of (1) magnitude/cost, (2) probability, and (3) coping potential separately. (1) Existing instruments for magnitude/cost appraisals have been mainly developed for clinical purposes, such as in research on panic or pain disorders, and therefore focus on negative appraisals, mainly catastrophizing [19,23]. We are not aware of an instrument focusing on positive magnitude/cost appraisals, that is, under- rather than over-estimations of threat outcomes. (2) Good instruments for the probability dimension exist (optimism questionnaires, e.g., LOT-R) [16,24]. (3) Available instruments for the coping dimension each focus on single sub-facets of high coping potential estimations, such as beliefs in controllability of events (e.g., Locus of Control) [25], beliefs in one's ability to act in a goal-conducive way (e.g., General Self Efficacy) [26], or perceptions of the availability of social support as a coping resource (e.g., Oslo Social Support Scale) [27]. For other sub-facets of coping potential, such as beliefs in one's ability to self-regulate or cope emotionally, or perceptions of other coping resources, such as financial assets, no validated instruments exist to our knowledge. More globally, there is no measure that combines all threat appraisal dimensions.

## Current paper

This overview motivates the development of new dedicated positive appraisal style self-report instruments, based on the process-content distinction. In the first approach, we develop a self-report questionnaire for the assessment of the frequency of cognitive processes relevant for a positive appraisal style (first approach: Perceived Positive Appraisal Style Scale, process-focused; short: PASS-process). For this purpose, we take items from existing, validated questionnaires that assess cognitive and behavioral strategies to deal with stressful situations, namely brief COPE and CERQ-short. An additional, self-generated subscale included in these analyses covers another important positive cognitive reappraisal tactic: distancing (also known as self-focused reappraisal, detachment, or decentering) [28–32]. We hypothesize that items with similar character on the dimensions of processes vs. contents, cognitive vs. behavioral, and positive vs. negative will cluster together, thus facilitating decisions about the inclusion of single items. We evaluate the resulting factors in terms of interpretability and theoretical compatibility and also apply statistical quality criteria, to eventually reach a solution that is satisfactory on all these levels.

The aim of the second part of this paper is to develop a scale that measures to what extent someone typically produces positive appraisals in stressful situations (Perceived Positive Appraisal Style Scale, content-focused; short: PASS-content). To this end as analyze responses to a new item pool in three independent German samples using factor analysis. Here, we only weakly hypothesize that the items will show a five-factor structure (three separate appraisal dimensions, cross-dimensional appraisals, appraisal background). The final solution considers interpretability, theoretical compatibility, and statistical quality.

The overarching aim of this study is to fill the gap in resilience research by providing two measures of self-perceived positive appraisal style, assessing appraisal processes and appraisal contents, respectively.

## Methods

### Participants

Analyses were based on data from three separate and independent samples of German participants, the Longitudinal Resilience Assessment (LORA; N = 1191) [33], the Gutenberg Brains Study (GBS; N = 3131) [34], and the Mainz Resilience Project (MARP; N = 202) [35–37].

**Table 1. Gender and age distribution of the LORA, MARP, and GBS samples used for the development of the scales.**

| Gender | Process-focused scale | | | | Content-focused scale | | | |
|---|---|---|---|---|---|---|---|---|
| | N (%) | Age, mean (*sd*) | Age, median (min, max) | Exclusions | N (%) | Age, mean (*sd*) | Age, median (min, max) | Exclusion |
| | | | | LORA | | | | |
| Male | 400 (34.5) | 29.1 (7.2) | 27 (18,50) | 6 | 400 (34.1) | 29.1 (7.4) | 27 (18,50) | 6 |
| Female | 753 (65.1) | 28.3 (8.2) | 26 (18,50) | 28 | 770 (65.5) | 28.3 (8.2) | 26 (18,50) | 11 |
| Total | 1157 (100) | 28.6 (7.9) | 26 (18,50) | 34 | 1,174 (100) | 28.6 (7.9) | 26 (18,50) | 17 |
| | | | | GBS | | | | |
| Male | 629 (36.9) | 43.4 (14.8) | 44 (18,75) | 522 | 595 (36.9) | 42.5 (14.2) | 42 (18,69) | 556 |
| Female | 1074 (63) | 40.1 (14.4) | 39(18,75) | 877 | 1,013 (62.9) | 40.0 (14.4) | 39 (18,69) | 938 |
| Total | 1704 (100) | 41.3 (14.6) | 41(18,75) | 1427 | 1,611 (100) | 40.9 (14.4) | 40 (18,69) | 1502 |
| | | | | MARP | | | | |
| Male | - | - | - | - | 93 (46.5) | 19.0 (0.8) | 19 (18,21) | 0 |
| Female | - | - | - | - | 105 (52.5) | 19.1 (0.8) | 19 (18,21) | 0 |
| Total | - | - | - | - | 200 (100) | 19.1 (0.8) | 19 (18,21) | 0 |

Some participants chose not to state their gender. Due to the low number of unknowns, their age is not reported separately, but they are included in the descriptive statistics of the total sample.

Recruitment lasted from February 2014 to March 2019 for the GBS sample and from 2016 to 2019 for the LORA and MARP studies. All participants took part in a screening interview before inclusion. Participants were mentally healthy at inclusion. Detailed descriptions of sampling and recruitment are published elsewhere. Descriptives of the samples are reported in Table 1. No data on race or ethnicity of participants was collected. All participants in all studies gave written informed consent, and the studies were approved by local ethics committees (Medical Board of Rhineland-Palatinate, Mainz, Germany, and the Ethics Committee of the Department of Medicine at the Goethe University Frankfurt am Main, Germany). Assessments were collected via paper-based or computer-based questionnaires. Authors only had access to pseudonymized data for the purposes of the analyses presented in this paper.

## Item pool

**Process-focused scale.** For appraisal processes, the majority of items were adopted from the COPE and CERQ questionnaires. The LORA and GBS studies, used for the development of a process-focused scale, employ the Brief COPE [21], which has 14 subscales composed of two items each. The CERQ has nine subscales composed of four (CERQ) or two (CERQ-short) items, respectively [14,20]. Both studies use the German version of the CERQ questionnaire that has 3 items per subscale [38]. Items and subscales were evaluated qualitatively by an expert panel to identify which subscales potentially capture positive appraisal processes. Subscales that could not be readily classified as there was ambiguity in regard to whether they target contents or processes, behaviors or cognitive processes, negative or positive appraisal processes (see Introduction), were retained initially. Nine Brief COPE subscales and four subscales of the CERQ questionnaire were excluded. Two self-generated items indexing distancing reappraisal (detachment, decentering) were also included in the further analysis. S1 Table gives an overview of excluded and included subscales using English-language example items. Prior to the factor analyses the data's skewness and kurtosis were evaluated in the LORA sample (see S2 and S3 Tables). This led to no exclusions.

**Content-focused scale.** For the development of a content-focused positive appraisal style scale, new German-language items were generated by experts in the field, see S6 Table. The reasoning underlying this decision was that other existing measures did not cover all dimensions of positive appraisal contents (see Introduction). Additionally, their inconsistent wording did not allow for comparable scores. The new items were intended to reflect appraisal tendencies on the three major threat appraisal dimensions (magnitude or cost, probability, and coping potential). Items were worded in easily understandable, non-technical language. Many were inspired by everyday language and statements as can be found in the self-help literature, in popular media reports, or in fiction dealing with themes of adversity, resilience, and coping. Realizing in the process of item generation that common language often does not differentiate the three appraisal dimensions, we also included items reflecting positive appraisal contents generally, across dimensions (e.g., 'I think that risks are often overestimated.', 'I usually see things in a negative fashion'). Further, it appeared that common language statements on how individuals evaluate threats often refer to more general attitudes or to an individual's perspectives on life, which in turn determine one's appraisal contents. Similarly, statements often invoke higher goals or purposes, which effectively provide positive appraisals, often in the form of global statements (e.g., 'I think that life is wonderful nevertheless', 'I trust in God', 'I think that even bad things make sense', 'I think one should not be upset by small things'). We thus included further items which we consider reflect a general appraisal background. We framed items both positively and negatively, to not prematurely exclude the possibility that low scores on negatively framed items might also express a positive appraisal style. In a second step, this pool was reduced based on theoretical and semantical discussions between the experts. The resulting item pool had 29 items. Response options were on a 4-point Likert scale, from "never" (1), "sometimes" (2), "often" (3), to "almost always" (4). Items were analyzed for their item-total correlation, item difficulty, skew, missings, and kurtosis in the LORA sample. Since the items had not been validated before, an additional sample (MARP) was used, to thereby ensure that findings would not be sample dependent. Items with an item-total correlation (part-whole corrected) of less than 0.3 or an item difficulty above 0.85 or below 0.2 were excluded. The factor analyses were conducted on the remaining items.

## Data cleaning

Only participants with complete data were considered, see Table 1. The completion rate in MARP was 100%, in LORA 98.6% and in GBS 95.7%. In the GBS sample the questionnaires were part of a later battery, leading to a lower number of participants in the analytical sample.

## Item selection

An exploratory factor analysis (EFA) was conducted separately for each approach (process and content focused) in the LORA sample, followed by a confirmatory factor analysis (CFA) in the GBS sample. All analyses were done separately for each sample and on item level. In the EFA, the items of the scales were submitted to Horn's parallel analyses [39] to identify the appropriate amount of factors. The scales use short Likert scales; therefore, the data was treated as ordinal [40]. Hence, an EFA with ordinary least squares estimation with polychoric correlations was conducted [41]. Oblimin rotation was applied. No continuity correction was applied, as the questionnaires had four or more answer options [42]. EFA results were evaluated based on model fit as well as the criterion to have at least two factors with at least five strongly loading indicators [43].

It is recommended that for a CFA in not-normally distributed data, a robust weighted least square estimator is used. Further, polychoric correlations were used. Model fit of CFA is

determined by a variety of indices, including the Tucker-Lewis Index (TLI; $\approx 0.95$), Comparative Fit Index (CFI; $\approx 0.95$), Root mean squared error (RMSEA; $< 0.06$), and Standardized Root Mean Squared Residual (SRMR; $< 0.08$). These indices were preferred over the Chi-squared statistic, as its significance is likely due to the sample size [44]. Additionally, the models were evaluated qualitatively, based on the wording and phrasing of the items, to evaluate the presence of potential method factors.

## Validation

The reliability and validity of the final two questionnaires were calculated with the GBS sample. Reliability was computed using McDonald's omega and Cronbach's alpha [45–47]. Convergent and discriminant validity were tested within the nomological network by evaluating Pearson's correlations of the scale score with measures of similar and dissimilar constructs [48]. The nomological network consists of a convergent measure such as optimism, which is a positive appraisal tendency on the probability dimension and known to be positively associated with mental health [49]. As such, optimism can be considered a parallel-level construct and expected to positively correlate with PAS. A further convergent measure is neuroticism, which is a propensity for negative emotionality and emotional dysregulation and a known vulnerability or risk factor [50]. Neuroticism was therefore hypothesized to negatively correlate with PAS. The network further includes a discriminant measure indicated by the concept of impulsive behavior, which is unrelated to positive appraisal and therefore expected to have no relationships with the PAS measure.

In the GBS sample, optimism is measured by the Optimism-Pessimism Short Scale-2 (SOP2) [51]. The scale consists of two items that are rated on a 7-point Likert scale. Neuroticism is measured by neuroticism subscale of the Big Five Inventory (BFI-10) [52], which includes two items rated on 5-point Likert scale. Impulsive behavior was assessed with the i8 scale for impulsive behavior [53]. The scale consists of four subscales, of which *Urgency* and S*pontaneity* measure impulsive behavior and *Intent* and *Perseverance* capture behavior in the opposite direction. We therefore only used the first two subscales within the nomological network. Furthermore, concurrent validity was assumed to be indicated by a moderate relationship with wellbeing, indicated by WHO-5 [54] as proxy of mental health.

All data, analysis code, and research materials are available by request. Data were analyzed using R version 4.1.2 using the psych [55] and lavaan [56] packages. This study's design and its analysis were not pre-registered.

## Results

### Perceived Positive Appraisal Style Scale, process-focused (PASS-process)

Item quality indices calculated in the LORA sample are reported in S2 and S3 Tables. No exclusions were made based on those results.

**Exploratory factor analysis in the LORA sample.** A parallel analysis using ordinary least squares estimation and polychoric correlations indicated that a total of nine factors should be retained, see S1 Fig. However, a closer inspection of the solutions showed that only solutions with less than five factors had at least two factors with five strongly loaded indicators. Therefore, the five-factor solution was further evaluated. Items that did not strongly load onto any factor were excluded one by one ($n = 5$ items). One item was from the CERQ-short *Putting into perspective* subscale, one item was from the self-generated *Distancing* subscale. The remaining three items were from the *Humor* and *Behavioral disengagement* subscales.

**Confirmatory factor analysis in the GBS sample.** The CFA used a robust weighted least squares estimation with polychoric correlations. The variances of the latent variables of factors

3, 4, and 5 were fixed due to the small numbers of indicators per factor. Factors 1 and 2 were highly correlated in the EFA, thus allowed to be correlated in the CFA model as well. The model fit was moderate to good (TLI = 0.94, NFI = 0.94, RMSEA = 0.094, SRMR = 0.078).

The factor analyses showed a structure that items that belong to the subscales *Positive reframing* (COPE), *Humor* (COPE), *Positive reappraisal* (CERQ), *Putting into perspective* (CERQ), and *Distancing* (self-generated) all strongly load on factor 1. Factor 2 was comprised of five items which belong to the subscales of *Acceptance* in both COPE and CERQ. Factors 3, 4, and 5 were each comprised of items of only one original subscale, respectively: *Other-blame* (CERQ), *Turning to religion* (COPE), and *Rumination* (CERQ), see Table 2. The first and second factors were substantially correlated ($r = 0.45$, $p < 0.000$).

**Interim discussion.** The item-level factor analyses showed a five-factor structure. Clustering was apparent insofar as the two first factors contained items originating from at least two different subscales and/or questionnaires and as these two factors were strongly correlated. This contrasted with factors 3 to 5, which all contained items originating from only one subscale and were not correlated to any other factor. Factor 1 and factor 2 included nearly all items from those subscales that we had a priori identified as the strongest candidates for reflecting processes (as opposed to contents), appraisals (as opposed to behaviors), and positive

**Table 2. Results from the CFA for the PASS-process.** Factor loadings by item.

| Item | Subscale (origin, Item number) | Factor 1 | Factor 2 | Factor 3 | Factor 4 | Factor 5 |
|---|---|---|---|---|---|---|
| I think that I can become a stronger person as a result of what has happened. | Positive reappraisal (CERQ-short, 6) | 0.63 | | | | |
| I think that the situation also has its positive sides. | Positive reappraisal (CERQ-short, 15) | 0.83 | | | | |
| I think I can learn something from the situation. | Positive reappraisal (CERQ-short, 24) | 0.85 | | | | |
| I think that it hasn't been too bad compared to other things. | Putting into perspective (CERQ-short, 16) | 0.68 | | | | |
| I tell myself that there are worse things in life. | Putting into perspective (CERQ-short, 25) | 0.64 | | | | |
| I try to look at the situation from an objective perspective. | Distancing | 0.48 | | | | |
| I have been trying to see it in a different light, to make it seem more positive. | Positive reframing (Brief COPE, 12) | 0.69 | | | | |
| I have been looking for something good in what is happening. | Positive reframing (Brief Cope, 17) | 0.71 | | | | |
| I have been making fun of the situation. | Humor (Brief COPE, 28) | 0.37 | | | | |
| I think that I have to accept that this has happened. | Acceptance (CERQ-short, 2) | | 0.74 | | | |
| I think that I have to accept the situation. | Acceptance (CERQ-short, 11) | | 0.85 | | | |
| I think I have to learn to live with the situation. | Acceptance (CERQ-short, 20) | | 0.67 | | | |
| I have been accepting the reality of the fact that it has happened. | Acceptance (Brief COPE, 20) | | 0.53 | | | |
| I have been learning to live with it. | Acceptance (Brief COPE, 24) | | 0.56 | | | |
| I feel that others are to blame for it. | Other Blame (CERQ-short, 9) | | | 0.84 | | |
| I feel that others are responsible for what happened. | Other Blame (CERQ-short, 18) | | | 0.90 | | |
| I feel that basically the cause lies with others. | Other Blame (CERQ-short, 27) | | | 0.69 | | |
| I have been trying to find comfort in my religion or spiritual beliefs. | Religion (Brief COPE, 22) | | | | 0.78 | |
| I have been praying or meditating. | Religion (Brief COPE, 27) | | | | 0.95 | |
| I often think about how I feel about what I have experienced. | Rumination (CERQ-short, 3) | | | | | 0.74 |
| I am preoccupied with what I think and feel about what I have experienced. | Rumination (CERQ-short, 12) | | | | | 0.56 |
| I want to understand why I feel the way I do about what I have experienced. | Rumination (CERQ-short, 21) | | | | | 0.85 |

English-language items correspond to the respective items of the German-language questionnaire versions used for instrument development.

(as opposed to negative) cognitions, namely *Positive reframing*, *Positive reappraisal*, *Putting into perspective*, *Distancing*, and *Acceptance*. These subscales provided a total of 13 of all 14 items loading on those factors. This constellation indicates that the two factors mainly capture the intended construct and strongly suggests that the item from the only other subscale providing loadings on these factors (*Humor*) should also be interpreted as targeting a positive appraisal process, rather than perhaps a distraction strategy (see Introduction). It can also be concluded from this result that the items loading onto factors 3 to 5 (*Other-blame*, *Turning to religion*, and *Rumination*) should be interpreted with caution and may well not be unambiguous cases of positive appraisal processes. We therefore decided to exclude them from the further questionnaire development.

Of the retained factors, factor 1 can be interpreted as indexing a self-perceived tendency to find appraisals that depict a *prima facie* difficult situation as relatively less negative, more positive, or less relevant. Factor 2 is restricted to one specific way of seeing a situation in a more positive light, by trying to find it acceptable. Hence, both factors are compatible with PASTOR's conceptualization of positive appraisal processes [6].

Although we computed the factor analyses on item level, the results show a clear loading of items based on the originally defined subscales. We therefore suggest that a pragmatic solution can be to restrict questioning to the corresponding two-item component subscales as frequently used in the Brief COPE and CERQ-short. Here, the *Positive reframing* and *Acceptance* subscales from the COPE can be dropped, given that they conceptually highly overlap with *Positive reappraisal* and *Acceptance* from the CERQ but loaded less on the respective factor. This leaves the five two-item subscales: *Humor* (COPE), *Acceptance* (CERQ), *Positive reappraisal* (CERQ), *Putting into perspective* (CERQ), and *Distancing* (self-generated). The resulting new scale of 10 items was next validated in the GBS sample.

**Validation.** Measurement reliability was very high, as indicated by McDonald's omega (ω = .85) and Cronbach's alpha (α = .78, 95% CI [.86, .87]). Moreover, the analyses indicated that reliability could not be significantly improved by dropping any single item, suggesting that the scale as a whole is consistent. This justified using the sum of all items for scoring in the validity analysis. Here, in the GBS sample, the scale's item sum score had a moderate positive correlation with optimism ($r$ = .27, 95% CI [.23,.32]) and a moderate negative correlation with neuroticism ($r$ = -.22, 95% CI [-.27,-.18]), indicating reasonable convergent validity. There was no relationship between the scale and *urgency* and *spontaneity* ($r$ = .1, 95% CI [.06,.15], $r$ = .05, 95% CI [.00,.09]), indicating discriminant validity. The scale had a moderate correlation with wellbeing ($r$ = .21, 95% CI [.17,.26]), indicating concurrent validity.

**Summary.** The present 10-item scale PASS-process has been collated from five different subscales originating from three different sources (COPE, CERQ, self-generated). Each subscale can be considered to represent a positive appraisal process that leads to a semantically distinct appraisal outcome, or content, including, for instance, an accepting, a relativizing, or a distanced perspective. In the item selection process, the corresponding items fell onto two different (though correlated) factors. Nevertheless, our reliability analyses showed high internal consistency of the resulting instrument combining the two factors. This indicates that the scale as a combination of the two factors measures the same latent construct. Accordingly, we name the instrument Perceived Positive Appraisal Style Scale, process-focused (PASS-process), and define that the PASS-process score consists of the sum of all items. PASS-process has sufficient convergent, discriminant, and concurrent validity. On this basis, it can be hypothesized that PASS-process will predict outcome-based resilience in longitudinal studies. PASS-process (see S5 Table) has been validated as a German-language instrument. For convenience, S4 Table gives the (unvalidated) English version.

## Perceived positive appraisal style scale, content-focused (PASS-content)

Based on the LORA and MARP samples, eight items were excluded from the pool of 29 items due to unsatisfactory item-total correlation and item difficulty. Item characteristics are reported in S6 Table. Reverse coded items were inverted prior to factor analysis.

**Exploratory factor analysis in the LORA sample.** A parallel analysis in LORA using ordinary least squares estimation and polychoric correlations suggested six factors, see S2 Fig. However, only a two-factor solution resulted in at least two factors with five strongly loading indicators. In this solution, we observed a clear grouping of items by valence and skew in both the MARP (pilot) and LORA samples. It was therefore concluded that the seven items with negative valence were a likely method factor, rather than a separate latent factor. Additional exploratory analyses were conducted on only the positively worded items, where no further factor structure with an adequate fit was found. Therefore, subsequent analyses continued with negative items all loading onto a single putative method factor and 14 positively worded items loading onto another single factor. These two factors were strongly inversely correlated ($r = -.75$).

**Confirmatory factor analysis in the GBS sample.** A confirmatory factor analysis was conducted with one factor consisting of the positively worded items and one putative method factor, see Table 3. A robust weighted least squares estimation with polychoric correlations was used. The resulting model had a good fit (TLI = 0.96, NFI = 0.96, RMSEA = 0.084, SRMR = 0.073).

**Interim discussion.** The (weakly) hypothesized five-factor structure, featuring positive appraisals on the three major threat appraisal dimensions as well as cross-dimensional positive

**Table 3. Factor loadings from the confirmatory factor model.**

| Item | Factor 1 | Factor 2* |
|---|---|---|
| I think that every difficult situation will end eventually. | 0.73 | |
| I think that I can deal successfully even with even the worst situation. | 0.69 | |
| I think that even bad things have a meaning. | 0.56 | |
| I think that it is better to assume a good ending if you don't know what is coming. | 0.56 | |
| I think that you should not be rattled by small things. | 0.58 | |
| I tend to see things rather optimistically. | 0.88 | |
| I think that there is a solution for every problem. | 0.77 | |
| I think that things will get better if you sit through them. | 0.4 | |
| I try to see things realistically, like they are. | 0.44 | |
| I think that you shouldn't make mountains out of molehills. | 0.59 | |
| For my goals and my ideals, I accept inconvenience. | 0.33 | |
| I think that I somehow always manage to get what I need. | 0.63 | |
| I think that things that initially seem bad often turn out well in the end. | 0.70 | |
| I think that life is wonderful after all. | 0.78 | |
| I tend to see things rather pessimistically. | | 0.90 |
| I fall into despair easily. | | 0.7 |
| I think that my needs are not satisfied. | | 0.57 |
| I take up a negative perspective. | | 0.85 |
| I have very little confidence in myself. | | 0.65 |
| I don't see any good aspects in negative experiences. | | 0.45 |
| I think that my goals are threatened. | | 0.55 |

*Items are negatively worded. Items are authors' translations of the original German-language items.

appraisals and positive appraisal background, was not found. Instead, the factor analyses showed two factors grouping all items of positive and negative valence, respectively. We interpret the negative valence factor as a method factor and exclude it from further analyses, since it is likely that the shared covariance do not represent a different construct, but rather shared semantics [43]. The positive valence factor can be interpreted as representing positive appraisal contents. All items with a loading of $> 0.3$ were retained [57]. The absence of further factor structure within the group of positively framed items may support the notion, developed during item generation, that common language may not readily distinguish the subtle appraisal-theoretical constructs on which the instrument development was based. On the basis of our results, it is also conceivable that threat appraisal is not structured along separable dimensions, but much more simply along a single good-bad dimension. We next validated the new 14-item questionnaire in the GBS sample.

## Validation

Measurement reliability, as indicated by McDonald's omega ($\omega = .88$) and Cronbach's alpha ($\alpha = .87$, 95% CI [.86, .87]), was very high, as expected given the results of the prior factor analysis. Subsequently, the total item sum score was used. The convergent validity of the new scale was high (optimism: $r = .58$, 95% CI [.55,.61], neuroticism: $r = -.41$, 95% CI [-.45,-.37]). The scale also had good discriminant validity: *urgency* ($r = .09$, 95% CI [.04,.14]), *spontaneity* ($r = -.06$, 95% CI [-.11,-.01]). Concurrent validity was moderate as indicated by the correlation with wellbeing ($r = -.46$, 95% CI [-.42,.5]). The scale had a substantial correlation with PASS-process ($r = .54$, 95% CI [.50,.60]).

   **Summary.**   We have generated a new 14-item questionnaire that enquires about the prevalence of positive appraisals in difficult situations. The instrument shows high reliability and consistency, indicating that a single latent construct is measured. Whether this indicates that there is only one dimension of threat appraisal (rather than the three dimensions assumed by PASTOR) or whether this reflects limitations of everyday language (which may often not be granular enough to differentiate between threat costs, probabilities, and coping) cannot be decided with the employed methods. The new questionnaire has good convergent, discriminant, and concurrent validity. We term it Perceived Positive Appraisal Style Scale, content-focused (PASS-content). It can be hypothesized that the scale will predict outcome-based resilience. See S7 Table for an English translation of the new instrument and S8 Table for the original German version.

## General discussion

The aim of the current work was to develop self-report measures for positive appraisal style (PAS) as defined by positive appraisal style theory of resilience (PASTOR) [4]. We generated two self-report questionnaires: (1) In a first approach, we developed a process-focused PAS questionnaire. To this end, we investigated the structure of the existing Brief COPE and CERQ-short instruments in combination with two self-generated items on distancing, to identify which of the coping and emotion regulation strategies covered therein could be classified as related to positive appraisal and reappraisal processes. The resulting questionnaire has ten items and is termed Perceived Positive Appraisal Style Scale, process-focused (PASS-process). (2) In a second approach, we developed a content-focused PAS scale for which a new item pool was created. We evaluated 29 self-generated items in a factor analysis with the intent to measure positive appraisal contents. This analysis resulted in a 14-item questionnaire, which we term Perceived Positive Appraisal Style Scale, content-focused (PASS-content).

## Process-focused scale

In the first approach on the PASS-process, a structure of five factors was present. The first factor was comprised by items of various clearly positive appraisal-related items. Well correlated with this factor was a second factor comprised of items targeting acceptance tactics. These two factors, which clustered together, appeared to be distinct from the other three factors, each uncorrelated with the first two factors and comprised of items from a single COPE or CERQ subscale only (*Other-blame*, *Turning to religion*, and *Rumination*). Initially, we speculated that these latter strategies could be also related to positive appraisal processes, rather than reflecting appraisal *contents*, *behavioral* coping strategies, or *negative* appraisal processes. While their apparent statistical separation from a cluster of items with unambiguous assignment does not exclude this possibility, it at least indicates that they may also reflect other constructs. We were thus not able to resolve the ambiguity around these items and preferred to not retain them. Retaining the subscales loading on the first two factors resulted in a highly internally consistent instrument, further confirming the observation of statistical clustering of the respective items.

From a theoretical point of view, the close statistical relationship between acceptance-based and non-acceptance-based items in the presence of separability as two factors is interesting. PASTOR employs a broad definition of positive reappraisal in particular as any cognitive change that modifies appraisals towards the better [6,10]. In this classification, trying to accept a difficult situation constitutes a sub-type of reappraisal, or specific reappraisal tactic, in which a situation that is first seen as so negative and annoying that needs to be changed is globally re-evaluated as a situation than can be managed by mentally and behaviorally adjusting to it. That is, an initial disappointing insight into one's inability to cope actively via situation modification (initial negative appraisal on the coping potential dimension) leads to an awareness of, and switch to, alternative coping possibilities (positive reappraisal on the coping potential dimension). There is an implicit undertone that the situation *will* be acceptable and not as disastrous as it might appear at first sight (positive reappraisal on the magnitude/cost dimension). PASTOR's broad classification of reappraisal, extending to acceptance, is grounded in the emotion process model [58]. This model recognizes only two classes of non-behavioral antecedents to emotional responses (namely attention assignment and appraisal) and, consequently, can accommodate only two ways of regulating emotional responses in an antecedent-focused manner (by redirecting attention or by changing the appraisal). Insofar as acceptance does not rely on attention modulation, it must be a way of changing appraisal, or re-appraising. The same classification has been employed in many studies [59–63]. In the clinical literature, however, acceptance is often also juxtaposed to reappraisal [12,64], although it is not stated where in the emotion process chain acceptance-based regulation could be situated.

We interpret our observation of a close statistical link between the tested acceptance and non-acceptance items as supporting the classification of acceptance as a reappraisal tactic, in line with our theoretical considerations. We conjecture that the observed factor separation is due to an easy semantic discriminability of acceptance-based processes from other types of positive appraisal and reappraisal processes. The latter focus not on one's ability to mentally and behaviorally adjust, but rather shift focus to elements of the situation that are not (as) disastrous. We also note that the brief COPE and CERQ use only two verbs to denominate acceptance ("accept"and "live with"), in contrast to the much more heterogeneous vocabulary used for the other positive (re)appraisal tactics.

Taken together, our results suggest PASS-process measures a single latent construct. From a practical point of view, the high internal consistency of the PASS-process permits sum scoring across all items of the scale. While it is usually not recommended to create sum scores of different factors, this can be done when the separate factor scores are understood to be indicating in the

same direction [65]. In the present case, higher scores in each factor can be understood to represent more frequent use of positive appraisal and reappraisal strategies in difficult situations.

Importantly, PASS-process is intended to measure process use (frequency or tendency) [66], not process effectiveness (ability or capacity). Experimental studies on emotion regulation suggest an association between the individual capacity to use reappraisal and the tendency to choose this strategy for emotion regulation [67]. Hence, PASS-process may be found to correlate with positive (re)appraisal effectiveness or capacity measures, such as employed in laboratory tasks [12]. One source of this hypothetical correlation may be a recall bias for pleasant success experiences. Another hypothetical source is a true relationship between frequency of use and effectiveness resulting from learning experiences by which processes that have been successfully used to appraise and reappraise aversive situations will be reinforced and more likely be recruited again in similar situations [68]. The existence of a use-effectiveness relationship, however, has to be tested in future studies.

### Content-focused scale

In the second approach on the PASS-content scale, we found a main factor loaded by the positively worded items. The three dimensions *magnitude* of threat, *probability* of threat and *coping potential*, as well as the added elements of cross-dimensional appraisal and appraisal background, were not found in the factor structure. We are hesitant to interpret this as evidence for the absence of the sub-facets of threat appraisal. Experimental studies show that the magnitude and the probability of an aversive outcome, such as a pain stimulus or loss of money, independently shape fear or stress reactions [69–72]. Likewise, experimental manipulations of control perceptions strongly influence aversive responses [73]. It is thus a reasonable possibility that our item pool did not contain enough statements clearly referring to single threat appraisal dimensions. This was likely a consequence of our deliberate effort to generate items that emulate everyday language, where subtle distinctions between aspects of threat may be less present than global, summarizing good-bad distinctions. A tendency for answering along a single good-bad dimension may also have been promoted by our inclusion of negatively framed items, in a sense that the presence of both positive and negative statements may have seduced participants to employ dichotomous thinking. This interpretation is supported by the finding of two anti-correlated factors grouping either positive or negative statements. On this basis, we conjecture that capturing different threat appraisal dimensions in a self-report tool would presumably require a larger and more differentiated item pool that contains no negatively framed items. Insofar as threat reactions are ultimately determined by an integration of appraisals along the various appraisal dimensions [6], such a more fine-grained development approach may, however, be of limited added value.

We interpret the factor comprised of all the negatively worded items as a method factor and excluded all negative items from the final scale. This is justified given its strong inverse correlation to the positive factor, meaning that it did not add further information. Also, in other studies, similar factors based on valence have been found [74–76]. This once more suggests that evaluative language may be strongly organized along a good-bad dimension. In addition, the specific context of filling in a questionnaire, requiring responding in short time to pre-formulated questions and excluding the possibility to formulate and elaborate on own statements, may well penalize more complex, non-dichotomous representations.

Given the instrument's consistency and its convergent, discriminant, and concurrent validity, the PASS-content sum score can be considered a good measure of one's perceived tendency to positively appraise difficult situations. As for PASS-process, PASS-content is limited to measuring consciously accessible and verbalizable mental contents, that is, PASS-content

does not inform about potential non-conscious appraisals. It therefore also suffers from content under-representation.

## Relation of PASS-process and PASS-content

The two questionnaires, PASS-process and PASS-content, measure two different, but related facets of PAS. Frequent use of positive appraisal processes, or strategies, in stressful circumstances (PASS-process) can be assumed to generate more positive stressor appraisals (PASS-content). This interpretation is supported by the high correlation between PASS-process and PASS-content observed in the GBS sample. Nevertheless, the two scales measure distinct constructs, rather than dimensions of the same construct. Firstly, not all use of positive appraisal or reappraisal strategies may lead to positive appraisal contents. This could be due to their ineffectiveness or the possibility that, despite the processes being effective, they do not generate new and consciously available contents measured by the PASS-content. Secondly, contents might be generated by processes not measured by the PASS-process. For example, appraisal contents could be generated in an automatic and unconscious way without the person becoming aware of the appraisal process used to generate them. Accordingly, the only partial correlation of PASS-process and PASS-content suggests that the scales may indeed be employed to assess at least partly separable aspects of the PAS construct.

## Limitations and future research

As stated, the two new scales, by being self-report instruments, are limited to consciously accessible and verbally reportable internal information. In addition, self-report instruments are limited by their inherent openness to bias, such as a tendency for socially desirable and internally consistent reporting, recall biases, framing effects, and mood congruency effects. With relevance to the appraisal topic, Koval et al. [77] have recently reported that the use frequency of different emotion regulation strategies, including positive cognitive reappraisal, reported with standard questionnaires is not correlated with their use frequency when measured in the same individuals with ambulatory methods in daily life. This points to the possibility that responding in retrospective self-report is strongly affected by self-models, or self-narratives, rather than veridically reflecting what individuals do or think in concrete situations. This does not exclude that PASS-process and PASS-content transport valuable information about appraisal styles. The appraisal of relevant life situations does not occur at one single moment, but repeatedly and also often retrospectively outside the concrete situation or event (when looking back at a challenging day or time of life, in communication with others, etc.) [78]. That is, appraisal must be understood as an integrating and longer-term process that, not only considers the immediate implications of a situation but also its more lasting consequences, its interactions with contextual factors, and its impact on overarching needs and goals (including a stable and positive identity) [6]. Hence, integrative appraisals are inextricably linked with a person's self-model, and a person's self-narrative about how one habitually appraises stressors can probably be understood as *being* that person's longer-term integrative appraisal tendency. Appraisal styles may thus be better addressed in a retrospective questionnaire than by an assessment of momentary reactions with ambulatory tools.

Nevertheless, Kalisch et al. [6] have emphasized the importance of developing complementary measurements of positive appraisal that do not require self-report. These would include behavioral tests, task-based functional imaging, physiological, or other bodily measures. A clear advantage of measuring perceived positive appraisal style with PASS-process and PASS-content is their suitability for large samples. Subsequent studies should address the open question which of the different possible routes to measuring appraisal will provide the best predictors for resilience.

A further limitation of the instruments is that both questionnaires use short Likert scales as response options. This has implications both for the assessment of the factor structure, as the data needed to be treated as ordinal [40], and for the variance captured by the scales. While this has been adjusted for in the factor analyses, when it comes to data collection, the scales are likely to only pick up on larger differences. Yet another potential limitation of the instruments is the use of sum scores. Sum scores for instruments built on factor solutions have recently come under criticism, as a sum score is a simpler model solution than the original factor solution [79]. However, it must also be considered that factor loadings are likely to vary from sample to sample and, thus, an equally weighted sum score is a more comparable indicator across studies [65]. Considering the good convergent, discriminant, and concurrent validity reported here for the PASS-process and PASS-content sum scores, the use of sum scores appears appropriate. Additionally, both scales do not include subscales, and comparisons between items or groups of items are not intended (though in theory possible between the five two-item subscales in PASS-process). Rather, the single sum score is meant to indicate a general self-perceived use of positive appraisal processes in the case of the PASS-process and the usual frequency of positive appraisal contents in the case of the PASS-content.

PAS, as measured by a previous version of the PASS-process when it was still under construction, has been shown to be associated with resilience in a cross-sectional sample of 16,000 individuals exposed to the first wave of the COVID-19 pandemic [17]. Future research will have to determine whether using the PAS scales that we introduce here presents any advantage over other instruments that only capture facets of positive appraisal, such as those mentioned in the introduction. That is, whether the PAS scales consistently outperform the related measures in comparative multivariate analyses such as using regularized regression techniques. This can be expected, given that PASS-content in particular was generated with the idea in mind that for those with a strong PAS, negative tendencies on one appraisal dimension can be compensated to some extent by positive tendencies on other dimensions.

The new scales should be employed in large longitudinal studies to investigate the role of PAS in resilience. Specifically, to address whether PAS is protective against specific dysfunctions or general mental health issues, as well as to investigate if it is protective in specific populations or specific kind of adverse circumstance, and less so in others. Future research will also have to determine the test-retest reliability and measurement invariance of the PAS scales and validate the PAS in different translations.

## Conclusions

In conclusion, appraisal is considered the key generative factor in stress responses and, consequently, in mental health and resilience [8,63,80]. The two scales presented in this study aim to measure a participant's perceived general tendency in how they appraise stressors. The PASS-process does so by measuring the processes or strategies someone employs to generate positive appraisals. The PASS-content measures the contents and thoughts that result from such processes. Both scales measure an individual's self-reported appraisal style. Although they do so by targeting related notions, we note that these are separate constructs. Being self-report instruments, these scales only capture the conscious and verbally accessible dimension of PAS. The interpretation of results should take this limitation into account.

## Supporting information

**S1 Fig. Parallel analysis for approach one, process-focused.**
(TIF)

**S2 Fig. Parallel analysis for approach one, content-focused.**
(TIF)

**S1 Table. Overview of subscales in the Brief COPE and CERQ-short and self-generated Distancing subscale.**
(DOCX)

**S2 Table. CERQ item evaluation.**
(DOCX)

**S3 Table. COPE item evaluation.**
(DOCX)

**S4 Table. Perceived positive appraisal style scale, process-focused.**
(DOCX)

**S5 Table. PASS-process German version.**
(DOCX)

**S6 Table. Evaluation of the items in the item pool for PAS.**
(DOCX)

**S7 Table. Perceived positive appraisal style scale, content-focused.**
(DOCX)

**S8 Table. PASS-content German version.**
(DOCX)

## Acknowledgments

We would like to thank all volunteers who took part in the study. We would also like to acknowledge the support of the administrative staff and students who helped in the success of the study.

## Author Contributions

**Conceptualization:** Oliver Tuescher, Ulrike Basten, Raffael Kalisch.

**Data curation:** Papoula Petri-Romão, Lara Puhlmann, Matthias Zerban, Rebecca J. Neumann, Aliaksandr Malyshau, Kira F. Ahrens, Bianca Kollmann.

**Formal analysis:** Papoula Petri-Romão, Haakon Engen, Anna Rupanova.

**Funding acquisition:** Michèle Wessa, Andreas Reif, Oliver Tuescher, Ulrike Basten.

**Investigation:** Rebecca J. Neumann, Aliaksandr Malyshau, Kira F. Ahrens, Anita Schick, Bianca Kollmann.

**Methodology:** Papoula Petri-Romão, Haakon Engen, Anna Rupanova, Andrea Chmitorz, Oliver Tuescher, Ulrike Basten, Raffael Kalisch.

**Project administration:** Papoula Petri-Romão, Anita Schick, Michèle Wessa, Michael M. Plichta, Andreas Reif, Ulrike Basten.

**Software:** Papoula Petri-Romão, Haakon Engen, Anna Rupanova.

**Supervision:** Michèle Wessa, Michael M. Plichta, Andreas Reif, Andrea Chmitorz, Oliver Tuescher, Ulrike Basten.

**Validation:** Papoula Petri-Romão, Raffael Kalisch.

**Visualization:** Papoula Petri-Romão.

**Writing – original draft:** Papoula Petri-Romão, Haakon Engen, Anna Rupanova, Raffael Kalisch.

**Writing – review & editing:** Papoula Petri-Romão, Haakon Engen, Anna Rupanova, Lara Puhlmann, Matthias Zerban, Rebecca J. Neumann, Aliaksandr Malyshau, Kira F. Ahrens, Anita Schick, Bianca Kollmann, Henrik Walker, Michael M. Plichta, Andreas Reif, Andrea Chmitorz, Oliver Tuescher, Ulrike Basten, Raffael Kalisch.

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
