## [Decision Letter · Decision Letter 0]

29 May 2023

PONE-D-23-08553Self-report assessment of Positive Appraisal Style (PAS): development of a process-focused and a content-focused questionnaire for use in mental health and resilience researchPLOS ONE

Dear Dr. Petri-Romão,

Thank you for submitting your manuscript to PLOS ONE. After careful consideration, we feel that it has merit but does not fully meet PLOS ONE’s publication criteria as it currently stands. Therefore, we invite you to submit a revised version of the manuscript that addresses the points raised during the review process. Please submit your revised manuscript by Jul 13 2023 11:59PM. If you will need more time than this to complete your revisions, please reply to this message or contact the journal office at plosone@plos.org. Please include the following items when submitting your revised manuscript:A rebuttal letter that responds to each point raised by the academic editor and reviewer(s). You should upload this letter as a separate file labeled 'Response to Reviewers'.A marked-up copy of your manuscript that highlights changes made to the original version. You should upload this as a separate file labeled 'Revised Manuscript with Track Changes'.An unmarked version of your revised paper without tracked changes. You should upload this as a separate file labeled 'Manuscript'.

We look forward to receiving your revised manuscript.

Kind regards,

Paulo Moreira, PhD

Academic Editor

PLOS ONE

Journal Requirements:

a) The name of the colleague or the details of the professional service that edited your manuscript.

b) A copy of your manuscript showing your changes by either highlighting them or using track changes (uploaded as a *supporting information* file).

c) A clean copy of the edited manuscript (uploaded as the new *manuscript* file).

"I have read the journal's policy and the authors of this manuscript have the following competing interests: RK has received advisory honoraria from JoyVentures, Herzlia, Israel. The other authors report no conflicts of interest." 

We note that you received funding from a commercial source: JoyVentures, Herzlia, Israel

Within this Competing Interests Statement, please confirm that this does not alter your adherence to all PLOS ONE policies on sharing data and materials by including the following statement: "This does not alter our adherence to PLOS ONE policies on sharing data and materials.” (as detailed online in our guide for authors http://journals.plos.org/plosone/s/competing-interests).  

If there are restrictions on sharing of data and/or materials, please state these. Please note that we cannot proceed with consideration of your article until this information has been declared. 

5. Please upload a new copy of Figure S1 and S2 as the detail is not clear. Please follow the link for more information:

https://blogs.plos.org/plos/2019/06/looking-good-tips-for-creating-your-plos-figures-graphics/

https://blogs.plos.org/plos/2019/06/looking-good-tips-for-creating-your-plos-figures-graphics/

**Additional Editor Comments:**

Reviewer 1 describes some issues that requires your attention, and I agree with Reviewer 1 about the need of adressing those issues in a revised version of your manuscript. 

Also, and from my own reading of the manuscript, my main concern refers to the construct validity, and the meaning of your proposed two independent scales to theory building.

For example, your aknowledge that "The two questionnaires, PASS-process and PASS-content, measure two different, but 703 related facets of PAS. Frequent use of positive appraisal processes, or strategies, in stressful 704 circumstances (PASS-process) can be assumed to generate more positive stressor appraisals 705 (PASS-content). This interpretation is supported by the high correlation between PASS- POSITIVE APPRAISAL STYLE SCALES 35 706 process and PASS-content observed in the GBS sample. Nevertheless, the two scales 707 measure distinct constructs".

On the other hand, you aknowledge that PASS-Process and PASS-Content are diferent dimensions, they are dimensions from the same construct of PASS. In fract, in pag. 29 you state that "Concurrent validity was moderate as indicated by 571 the correlation with wellbeing (r = -.46, 95% CI [-.42,.5]). The scale had a substantial 572 correlation with PASS-process (r = .54, 95% CI [.50,.60])." What the correlation of .54 tells us about these 2 dimensions? 

So, did you considerer a model where you would have two higher order correlated factors (PASS-Content and PASS-Process)?  

I would invite you to expand and elaborate on this issue, as it would help the readers to better undertand the meaning of these psychometric results to theory building. 

In insure you that this decision is based on the PLOS ONE's publication criteria and not in others such as novelty or perceived impact. 

Reviewers' comments:

Reviewer's Responses to Questions

**Comments to the Author**

1. Is the manuscript technically sound, and do the data support the conclusions?

Reviewer #1: Yes

2. Has the statistical analysis been performed appropriately and rigorously? 

Reviewer #1: Yes

3. Have the authors made all data underlying the findings in their manuscript fully available?

Reviewer #1: Yes

4. Is the manuscript presented in an intelligible fashion and written in standard English?

Reviewer #1: Yes

5. Review Comments to the Author

Reviewer #1: Summary

The study presents valuable contributions to the field by developing two self-report questionnaires, PASS-process and PASS-content, which measure individual appraisal and coping with stressors and positive appraisal contents, respectively. With good construct validity and internal consistency, the scales demonstrate potential for various mental health and resilience research applications.

Introduction

- The introduction is informative and provides a comprehensive background. However, it could benefit from some streamlining. The word count is considerable. The authors should consider simplifying details and summarizing non-essential parts, maintaining direct relevance to the PAS scales. For example, the definition of resilience (page 3, paragraph 1) and appraisal styles and examples (page 3, paragraph 2) could be summarized and shortened. Similarly, details before the subheading of Positive appraisal style and resilience on page 5 appear less relevant to the main topic and may be summarized or removed.

- There are many instances where statements are made without supporting references. This lapse in citation might leave readers uncertain about the origins of these assertions. For instance, on page 4, paragraph 2, it is unclear which previous studies or papers support the three main reasons for PAS being an "interesting construct for resilience research".

- On page 6, line 129, the term "general need for more informative self-report instrument" is vague. A clearer explanation would be helpful, including how this new instrument is more informative than previous ones and whether there might be potential disadvantages, such as increased dropout rates for longer instruments.

- Contexts about the details of approach implemented in the study in the introduction can be better placed in the method section.

Methods

- The methods section covers the analysis of items from existing questionnaires (Brief COPE and CERQ-short) for PASS-process development and the creation of a new item pool for PASS-content. According to page 12, paragraph 1, there are existing measures of certain dimensions of positive appraisal. What motivated the decision to create a new pool of items rather than leveraging these existing measures, as was done for the PASS-process scale?

- On page 19, line 404, the authors state, "Additionally, the models were evaluated qualitatively", without specifying the methods and criteria used.

Results and Data

- In general, the results are well presented. However, the completion rate for each sample was not reported (page 17, line 380). I would suggest the authors to specify the percentage of participants in each sample whose data was included as complete.

- The reason for excluding participants who did not specify their gender should be clarified (Page 18, Table 1).

- On page 23, table 2: is CERQ-COPE a typo? And is it correct that the last 8 items showed identical loading coefficients to other items belonging to the same factor? For instance, the loadings of the Rumination items (CERQ-short, 3, 12 and 21) are all 0.73.

- On page 23, Table 2, there might be a typo: "CERQ-COPE." Additionally, it appears that the last eight items showed identical loading coefficients to other items belonging to the same factor, which is unusual. To illustrate, the Rumination items from the CERQ-short (items 3, 12, and 21) all showed a factor loading of 0.73. Is this indeed accurate, or could there have been a potential error in the reported values?

- Please specify the reason to drop all negatively worded items in the PASS-content scale as the CFA model fit is acceptable. Interestingly, the author chose to keep items with relatively low loading (e.g. "For my goals and my ideals, I accept inconvenience. loading = 0.33)" in the final version of the scale.

- Minor errors in the supporting material, such as the phrase "approach one" on page 48, line 1021, need to be corrected to "approach two". The term "approach one and two" are not used consistently throughout the manuscript.

6. PLOS authors have the option to publish the peer review history of their article (what does this mean?). If published, this will include your full peer review and any attached files.

Reviewer #1: No

---

## [Author Response · Author response to Decision Letter 0]

15 Nov 2023

1. Editor: Please ensure that your manuscript meets PLOS ONE's style requirements, including those for file naming. The PLOS ONE style templates can be found at https://journals.plos.org/plosone/s/file?id=wjVg/PLOSOne_formatting_sample_main_body.pdf and https://journals.plos.org/plosone/s/file?id=ba62/PLOSOne_formatting_sample_title_authors_affiliations.pdf

Reply: We have carefully revised the style requirements and made the following adjustments:

The supplementary figure files have been renamed. Figures in text are now referred to as “Fig”. 

The format of the tables has been adjusted to include normal gridlines. The format of the table titles has also been adjusted. 

Supplementary tables and figures are now referenced in text correctly (e.g. S1 Fig, rather than Fig S1). File names have been adjusted accordingly. 

2. Editor: We suggest you thoroughly copyedit your manuscript for language usage, spelling, and grammar.

a) The name of the colleague or the details of the professional service that edited your manuscript.

b) A copy of your manuscript showing your changes by either highlighting them or using track changes (uploaded as a *supporting information* file).

c) A clean copy of the edited manuscript (uploaded as the new *manuscript* file).

Reply: Upon your suggestion we have requested the help of our colleague India Sawyer to edit the language usage, spelling and grammar. The edits are visible in the track changes file and included in the clean copy of the revised manuscript. In general, some sentenced have been shortened for increased readability and redundant wordings have been reduced. 

3. Editor: We note that the grant information you provided in the ‘Funding Information’ and ‘Financial Disclosure’ sections do not match. 

Reply: The numbers have been carefully checked. The information “subproject Z03” has been added to the Award of the Deutsche Forschungsgemeinschaft CRC 1193 in the Funding Information section.

4. Editor: Thank you for stating the following in the Competing Interests section: 

"I have read the journal's policy and the authors of this manuscript have the following competing interests: RK has received advisory honoraria from JoyVentures, Herzlia, Israel. The other authors report no conflicts of interest." 

We note that you received funding from a commercial source: JoyVentures, Herzlia, Israel

Within this Competing Interests Statement, please confirm that this does not alter your adherence to all PLOS ONE policies on sharing data and materials by including the following statement: "This does not alter our adherence to PLOS ONE policies on sharing data and materials.” (as detailed online in our guide for authors http://journals.plos.org/plosone/s/competing-interests). 

If there are restrictions on sharing of data and/or materials, please state these. Please note that we cannot proceed with consideration of your article until this information has been declared. Please include your amended Competing Interests Statement within your cover letter. We will change the online submission form on your behalf.

Reply: We apologize for not being precise in our declaration of potential competing interests in the previous version of the manuscript. The receipt of advisory honoraria by author RK from the commercial source JoyVentures, Herzlia, Israel does not, in our eyes, interfere with the full and objective conduct and presentation of the research. We have declared the receipt of honoraria because we cannot exclude that this activity might be perceived by others as a potentially competing interest. Thus, our intention was to be fully transparent.

To align our competing interest statement with the journal guideline, we now say: RK has received advisory honoraria for consultancy from the commercial source JoyVentures, Herzlia, Israel. This does not interfere with the full and objective conduct and presentation of the research and does not alter our adherence to PLOS ONE policies on sharing data and materials.

5. Editor: Please upload a new copy of Figure S1 and S2 as the detail is not clear. Please follow the link for more information: https://blogs.plos.org/plos/2019/06/looking-good-tips-for-creating-your-plos-figures-graphics/
https://blogs.plos.org/plos/2019/06/looking-good-tips-for-creating-your-plos-figures-graphics/

Reply: The supplementary figures have been recreated with a dpi of 600 and uploaded in the correct format (S1_Fig.tif, S2_Fig.tif). 

6. Editor: Reviewer 1 describes some issues that requires your attention, and I agree with Reviewer 1 about the need of adressing those issues in a revised version of your manuscript. 

Also, and from my own reading of the manuscript, my main concern refers to the construct validity, and the meaning of your proposed two independent scales to theory building.

For example, your aknowledge that "The two questionnaires, PASS-process and PASS-content, measure two different, but 703 related facets of PAS. Frequent use of positive appraisal processes, or strategies, in stressful 704 circumstances (PASS-process) can be assumed to generate more positive stressor appraisals 705 (PASS-content). This interpretation is supported by the high correlation between PASS- POSITIVE APPRAISAL STYLE SCALES 35 706 process and PASS-content observed in the GBS sample. Nevertheless, the two scales 707 measure distinct constructs".

On the other hand, you aknowledge that PASS-Process and PASS-Content are diferent dimensions, they are dimensions from the same construct of PASS. In fract, in pag. 29 you state that "Concurrent validity was moderate as indicated by 571 the correlation with wellbeing (r = -.46, 95% CI [-.42,.5]). The scale had a substantial 572 correlation with PASS-process (r = .54, 95% CI [.50,.60])." What the correlation of .54 tells us about these 2 dimensions? 

So, did you considerer a model where you would have two higher order correlated factors (PASS-Content and PASS-Process)? I would invite you to expand and elaborate on this issue, as it would help the readers to better undertand the meaning of these psychometric results to theory building. 

In insure you that this decision is based on the PLOS ONE's publication criteria and not in others such as novelty or perceived impact.

Reply: We appreciate this issue being brought to our attention, as we tried to make clear that appraisal processes and appraisal content are two distinct constructs, rather than dimensions of the same concept. The two scales are therefore two complimentary strategies of assessing PAS. The high correlation between the two scales is expected, since appraisal processes are necessary to generate appraisal contents. The PASS-process scale assesses how frequently strategies that could lead to positive appraisals are used. Therefore the PASS-content scale assesses how often a person has thoughts that denote positive appraisals, independent of what strategies were used. It is however likely, that frequent use of strategies will lead to more positive appraisal thoughts. We would therefore expect that the scales are correlated. However, since not all appraisal processes are assessed with the PASS-process scale and frequent use will not always mean that the processes generate appraisal thoughts. Therefore, the correlation should not be 1 between the two scales. In the manuscript we discuss these questions in the ‘Discussion’ section in the subheading ‘Relation of PASS-process and PASS-content’. We have further included explanations in the introduction to prime the reader to our results and justify our approaches (“Appraisal processes versus appraisal contents”, 113-117; “Developing a PAS measurement”, 198-201)

The PASS-content can be understood as a more direct measure of PAS. The added value of the PASS-process is specifically interesting to researchers focused on the strategies used to generate the contents. Alternatively, both measures can be used to get an insight into both facets of PAS, as is the case in the DynaMORE project (EU Horizon 2020 777084).

Introduction

Reviewer: The introduction is informative and provides a comprehensive background. However, it could benefit from some streamlining. The word count is considerable. The authors should consider simplifying details and summarizing non-essential parts, maintaining direct relevance to the PAS scales. For example, the definition of resilience (page 3, paragraph 1) and appraisal styles and examples (page 3, paragraph 2) could be summarized and shortened. Similarly, details before the subheading of Positive appraisal style and resilience on page 5 appear less relevant to the main topic and may be summarized or removed.

Reply: We agree that the introduction can be streamlined and have made changes accordingly. We have significantly reduced the length of the first subsection, as well as the subsections ‘Positive appraisal style and resilience’, ‘Current paper’. Further edits have been made to the remaining subsections to improve reader flow. We have retained parts that we felt were important to justify the motivation of this work and introduce the reader to the concept. The introduction has been shortened by 2 pages (approx. 50 lines). 

Reviewer: There are many instances where statements are made without supporting references. This lapse in citation might leave readers uncertain about the origins of these assertions. For instance, on page 4, paragraph 2, it is unclear which previous studies or papers support the three main reasons for PAS being an "interesting construct for resilience research".

Reply: References have been added to justify the statement “interesting construct for resilience research”. Further references have been added in lines 58 & 141.

Reviewer: On page 6, line 129, the term "general need for more informative self-report instrument" is vague. A clearer explanation would be helpful, including how this new instrument is more informative than previous ones and whether there might be potential disadvantages, such as increased dropout rates for longer instruments.

Reply: We have removed this statement, as we significantly reduced the length of this section. In an earlier paragraph, in line 62 we previously included a reference to explain the reason why self-report measures are needed, which we felt was sufficient. 

Reviewer: Contexts about the details of approach implemented in the study in the introduction can be better placed in the method section.

Reply: We have placed most part of the ‘Current paper’ section in the methods section and removed parts that were redundant.

Methods

Reviewer: The methods section covers the analysis of items from existing questionnaires (Brief COPE and CERQ-short) for PASS-process development and the creation of a new item pool for PASS-content. According to page 12, paragraph 1, there are existing measures of certain dimensions of positive appraisal. What motivated the decision to create a new pool of items rather than leveraging these existing measures, as was done for the PASS-process scale?

Reply: The CERQ and COPE cover a large amount of appraisal processes. In contrast, the shortcomings of existing measures of PAS contents at detailed in lines 242 to 257. Some elements of some dimensions are covered by existing scales. There are no measures that included extensive facets of coping potential or positive aspects of cost appraisals. Specifically though, the existing measures are not phrased consistently in relation to facing difficult situations, therefore not allowing direct comparisons and joint analyses of answers. 

This clarification has been added to the text (line 328).

Reviewer: On page 19, line 404, the authors state, "Additionally, the models were evaluated qualitatively", without specifying the methods and criteria used.

Reply: We have now added the following clarification to the sentence:

The models were evaluated qualitatively, based on the wording and phrasing of the items, to evaluate the presence of potential method factors (line 379).

Results and Data

Reviewer: In general, the results are well presented. However, the completion rate for each sample was not reported (page 17, line 380). I would suggest the authors to specify the percentage of participants in each sample whose data was included as complete.

Reply: The completion rate has been added in line 358. The completion rate in MARP was 100 %, in LORA 98.6 % and in GBS 95.7 %.

Reviewer: The reason for excluding participants who did not specify their gender should be clarified (Page 18, Table 1).

Reply: People were not excluded based on gender, simply, the number of participants with unspecified gender were not reported in the table, since the mean would be based on =< 4 participants. This has been clarified.

Reviewer: On page 23, table 2: is CERQ-COPE a typo? And is it correct that the last 8 items showed identical loading coefficients to other items belonging to the same factor? For instance, the loadings of the Rumination items (CERQ-short, 3, 12 and 21) are all 0.73.

Reply: We apologize for these errors. An incorrect Table 2 had been included in the manuscript. The correct table without typos and with correct factor loadings has now been placed in the text. 

Reviewer: Please specify the reason to drop all negatively worded items in the PASS-content scale as the CFA model fit is acceptable. Interestingly, the author chose to keep items with relatively low loading (e.g. "For my goals and my ideals, I accept inconvenience. loading = 0.33)" in the final version of the scale.

Reply: In regard to the negatively worded items we follow the recommendation by Rosellini and Brown (2021). The fact that all negatively worded items strongly load onto one factor (rather than for example, negatively loading onto factor 1) strongly indicates that shared covariance does not represent a different latent construct, but rather shared semantics. It would therefore not be justified to include these items in a score. They could have been retained in order to detect inattention, but one of the stated aims of the scales was to have a short scale that could be applied to large samples and would not increase the burden on participants unduly.

Following the literature (e.g. Brown, 2015) we applied a pre-defined threshold of 0.3-0.4 for the retention of items. On this basis, the items with a loading of 0.33 were retained.

Clarifying statements have been added to the interim discussion (527-530).

Reviewer: Minor errors in the supporting material, such as the phrase "approach one" on page 48, line 1021, need to be corrected to "approach two". The term "approach one and two" are not used consistently throughout the manuscript.

Reply: These errors have been fixed.

---

## [Editor Report · Decision Letter 1]

27 Nov 2023

Self-report assessment of Positive Appraisal Style (PAS): development of a process-focused and a content-focused questionnaire for use in mental health and resilience research

PONE-D-23-08553R1

Dear Dr. Petri-Romão,

We’re pleased to inform you that your manuscript has been judged scientifically suitable for publication and will be formally accepted for publication once it meets all outstanding technical requirements.

Kind regards,

Paulo Alexandre Soares Moreira, PhD

Academic Editor

PLOS ONE
---

## [Editor Report · Acceptance letter]

25 Jan 2024

PONE-D-23-08553R1 

PLOS ONE

Dear Dr. Petri-Romão, 

I'm pleased to inform you that your manuscript has been deemed suitable for publication in PLOS ONE. Congratulations! Your manuscript is now being handed over to our production team.

Kind regards, 

on behalf of

Professor Paulo Alexandre Soares Moreira 

Academic Editor

PLOS ONE